# A Design Methodology for Fault-Tolerant Neuromorphic Computing Using Bayesian Neural Network

**DOI:** 10.3390/mi14101840

**Published:** 2023-09-27

**Authors:** Di Gao, Xiaoru Xie, Dongxu Wei

**Affiliations:** 1The School of Intelligent Manufacturing, Hangzhou Polytechnic, Hangzhou 311402, China; gaodi1995@outlook.com; 2The School of Electronic Science and Engineering, Nanjing University, Nanjing 210023, China; 3The College of Information Science and Electronic Engineering, Zhejiang University, Hangzhou 310027, China

**Keywords:** neuromorphic computing, memristor crossbar array, process variation, Bayesian neural network, variational inference

## Abstract

Memristor crossbar arrays are a promising platform for neuromorphic computing. In practical scenarios, the synapse weights represented by the memristors for the underlying system are subject to process variations, in which the programmed weight when read out for inference is no longer deterministic but a stochastic distribution. It is therefore highly desired to learn the weight distribution accounting for process variations, to ensure the same inference performance in memristor crossbar arrays as the design value. In this paper, we introduce a design methodology for fault-tolerant neuromorphic computing using a Bayesian neural network, which combines the variational Bayesian inference technique with a fault-aware variational posterior distribution. The proposed framework based on Bayesian inference incorporates the impacts of memristor deviations into algorithmic training, where the weight distributions of neural networks are optimized to accommodate uncertainties and minimize inference degradation. The experimental results confirm the capability of the proposed methodology to tolerate both process variations and noise, while achieving more robust computing in memristor crossbar arrays.

## 1. Introduction

Neuromorphic computing represents an exciting and promising approach to artificial intelligence (AI) that models its elements on the human brain and nervous system. Instead of relying on a central processing unit (CPU) like traditional von Neumann architectures, neuromorphic systems distribute computations across interconnected nodes, inspired by the remarkable capabilities of the human brain. This parallelism enables efficient and real-time processing of data, making it well suited to tasks such as image and speech recognition [1,2]. Recent research in the field of non-volatile memory technologies has enabled memory devices to tune their conductance and store multi-level states [3,4] such as resistive RAM (ReRAM), etc. On this basis, the utilization of memristor crossbar arrays to speed up matrix–vector multiplication (MVM) operations holds great promise for building energy-efficient and high-performance neural network architectures.

However, the operations of a memristor crossbar array are indeed susceptible to various non-ideal effects due to its analog computing nature. These non-ideal effects can arise from several aspects, such as driver/sensing resistance, analog-to-digital converter (ADC) and digital-to-analog converter (DACs) non-linearity, process variations, and other physical phenomena. The above-mentioned faults can be further categorized to the device or circuit level [5,6].

This paper mainly focuses on device-level fault sources, including process variations [7,8], analog fluctuations, and noise during read and write operations [3,9,10]. The stochastic behavior exhibited by memristor devices can introduce deviations in its conductance, which then inevitably affect the reliability of the computations performed on the whole memristor crossbar array. Researchers are actively exploring multiple approaches to enhance the overall performance of memristor-based analog computing systems, such as redundancy and error correction [11], adaptive algorithms [12], statistical analysis and modeling [13], and even their combination.

Process variations are inherent in nano-technology manufacturing processes, and can be decomposed into global and local variations. Global components are shared by all the memristors on a die. Local variation is related to the proximity effect and results in variation correlation of the devices across the array. Their combined impact makes neural network parameters heavily deviate from their expected values, thus making the hardware inference performance also significantly deviate from the software validation accuracy. As shown in Figure 1, when mapping a neural network model onto one or more memristor crossbar arrays, the weights of the model are typically represented by the conductance values of the memristors. However, the programmed weights stored in the memristors may deviate from the expected values, i.e., the actual synapse weight may follow a distribution instead of a point value; thus, the inference accuracy cannot reach the same level as in training.

From a system designer’s perspective, an as good as possible inference performance is always desired to maximize the benefit of the underlying system. Note that improving the fault tolerance of memristor crossbar arrays is an ongoing research area. Advancements in device fabrication, circuit design, and algorithmic techniques are necessary to develop effective solutions that mitigate the impact of conductance variations and enhance the fault tolerance of memristor-based neuromorphic computing systems.

In this work, we propose a novel design methodology for fault-tolerant neuromorphic computing to address the weight disturbance problem without involving either redundancy design or adaptive retraining/remapping. In particular, by leveraging the probabilistic nature of Bayesian inference, we employ a Bayesian neural network (BNN) architecture to adjust the synapse weights of memristors, accounting for process variations. Specifically, we first model the synapse weights as probability distributions considering the effects of both process variations and noise, i.e., the programmed memristor conductances follow a log-normal distribution [12]. On this basis, we introduce a priori from the same distribution family to estimate the optimal posterior distributions over network parameters that can accommodate uncertain and stochastic behavior in memristor conductance.

## 2. Background and Related Work

A memristor is a two-terminal electronic device for which the resistance changes with the voltage applied across it, allowing for non-volatile storage and information representation [14,15,16,17]. The memristor mentioned in this paper uses resistive random-access memory (ReRAM), but the proposed methodology is also compatible with other memristor technologies.

Figure 2 shows an overview of a memristor-based neuromorphic computing system as well as the matrix–vector multiplication (MVM) operation conducted on memristor crossbars. In particular, a memristor crossbar array of M×N is programmed to represent the neural network weights by tuning the conductances [1,18]. When applying a input voltage vector *V* = [$V_{1}$, $V_{2}$,…, $V_{M}$] to the wordlines, the array can output a current vector *I* = [$I_{1}$, $I_{2}$,…, $I_{N}$] according to Kirchhoff’s laws. Benefiting from the structure of crossbar arrays, multiple dot products can be computed simultaneously in a single operation [19]. This parallelism enables efficient and high-throughput MVM operations. Additionally, the non-volatile nature of ReRAM allows for the retention of weight values even when the power is turned off, further eliminating the need for frequent weight reloading while reducing the overhead associated with data transfer.

According to prior work [5,6], the error sources in memristor crossbar arrays include the following: (i) non-ideal peripheral circuitry (DACs and ADCs); (ii) circuit non-idealities (wire resistance, sneak paths, source and sink resistances); and (iii) device deviations due to process variations and thermal, drift, and limited endurance. Considering the analog nature of memristor crossbar arrays, the computational correctness of the underlying MVM operations is easily compromised due to deviations or errors. Note that the inherent presence of process variations and noises makes the weights represented by the synaptic elements within the crossbar array always stochastic. Thus, in the following, we will focus on the effects of process variations and investigate the corresponding fault-tolerant technologies.

Generally speaking, process variations are inherent in the nano-technology manufacturing process [7,8]. Existing work on memristor variation characterization found that it follows a log-normal distribution after the collective motion of many atoms in changing the state of a memristor [12]. To mitigate the impacts of devices with high variations, many adaptive algorithms such as retraining and remapping have been applied in prior work [12,14]. However, both the retraining and remapping cannot depart from the support of precise variation measurements at a post-silicon stage, which consumes huge testing costs and hardware overheads. Prior work [20,21] also employed a Bayesian optimization method to improve the robustness of analog DNNs, while presenting a thorough theoretical study about the interactions between fault tolerance and the neural network architecture choice (i.e., dropout, normalization, etc.) as well as its influence on the robustness. In contrast, Bayesian optimization in the above work [20,21] was used to automatically search for the optimal neural network architecture which would be robust to weight drifting, while the objective of our proposed BNN method is to learn the true posterior distribution presented by the memristor conductance states.

## 3. Preliminary Work on Bayesian Neural Networks

In this section, we give a brief introduction to Bayesian neural networks. Bayesian neural networks (BNNs) combine the principles of neural networks with the Bayesian inference technique, offering a probabilistic framework for modeling and learning from data. Unlike traditional neural networks that use fixed weights, BNNs assign probability distributions to the weights, thus allowing for uncertainty estimation in predictions and more robust decision making.

Specifically, the training process of a traditional NN model is viewed as performing a maximum likelihood estimation (MLE) [22,23]. The goal of MLE is to find a set of network parameters that maximizes the likelihood of observing the given data:(1)$$w_{\mathrm{MLE}}={\mathrm{argmin}}_{w}-\log P(D|w)$$
in which P(D|w) is the likelihood, representing the probability of observing the training data D given the model’s parameters w.

In contrast to MLE, which aims to find a set of point estimates of the optimal parameters, Bayesian inference directly computes the probability distribution of the weights that is also known as posterior:(2)P(w|D)=P(w)P(D|w)/P(D)
in which the prior distribution P(D|w) captures the initial beliefs about the weights before observing any data, reflecting the assumptions or knowledge about the weights’ likely values.

Bayesian neural networks combine the prior distribution with the likelihood of the data given the weights to compute the posterior distribution P(w|D). The posterior distribution provides a more comprehensive representation of uncertainty compared to a single point estimate obtained through MLE. Instead of only finding the optimal point estimate of the weights, the posterior distribution represents the entire range of plausible values for the weights, along with their associated probabilities. In general, various techniques can be employed to compute the posterior distribution, such as Markov Chain Monte Carlo (MCMC) methods [24], variational inference [25,26], or other approximate inference techniques [27]. During the prediction phase, BNNs can use the posterior distribution to generate a distribution of predictions, thereby making more robust decisions in applications where uncertainty is critical.

To make a prediction given a input sample $x^{*}$, a traditional NN with deterministic weights w predicts the output as $P(y^{*}|x^{*},w)$, while a BNN considers the weights as variational and makes the prediction by computing the expected value over the posterior distribution P(w|D): $P\left(y^{*}|x^{*}\right)=E_{P(w|D)}\left[P\left(y^{*}|x^{*},w\right)\right]$. To address the intractable computation of the posterior distribution, previous research [24,26] suggested finding a variational approximation to the posterior on the weights, also known as variational inference.

Finding the variational distribution that is closest to the true posterior distribution corresponds to minimizing the Kullback–Leibler (KL) divergence between two probability distributions. In the context of BNNs, the KL divergence is used to quantify the discrepancy between the posterior distribution and the variational distribution. Mathematically, the learning process of BNNs can be formulated as minimizing the following objective function given the vatiational parameters θ:(3)$$\begin{array}{cc} \theta^{*} & ={\mathrm{argmin}}_{\theta }\mathrm{KL}\left[q\left(w|\theta \right)||P\left(w|D\right)\right]\\ & ={\mathrm{argmin}}_{\theta }\int \log\frac{q(w|\theta )}{P(w)P(D|w)}dw\\ & ={\mathrm{argmin}}_{\theta }\mathrm{KL}\left[q\left(w|\theta \right)||P\left(w\right)\right]-E_{q(w|\theta )}\left[\log P\left(D|w\right)\right]+\log P\left(D\right) \end{array}$$

Combined with the maximum likelihood estimation observing the data, the objective function of a Bayesian neural network model can be expressed as:(4)$$F\left(D,\theta \right)=\mathrm{KL}\left[q\left(w|\theta \right)||P\left(w\right)\right]-E_{q(w|\theta )}\left[\log P\left(D|w\right)\right]=-\mathrm{ELBO}$$

The above objective function is a sum of a data-dependent part (the former) referred to as the likelihood cost, and a prior-dependent part (the latter) referred to as the complexity cost. Taking the negative sign of the resulting objective function is known as the variational evidence lower bound (evidence lower bound, ELBO) [23]. The learning process is a trade-off between fitting the data D and satisfying the prior P(w), subtly combining the flexibility and learning capabilities of neural networks with the probabilistic framework of Bayesian inference.

## 4. Proposed Methodology

### 4.1. Overview

To overcome the challenges of memristor deviations, we investigate and propose a design methodology for fault-tolerant neuromorphic computing using a Bayesian neural network. The proposed methodology mainly focuses on the faults incurred by process variations and noise at the device level, assuming other components and operations are error-free. Process variation refers to the inherent differences that occur during the fabrication of electronic components. Combined with the random fluctuations in the conductance values that occur during read and write operations, the weights involved in the forward propagation inevitably deviate from the offline trained values. These deviations can impact the performance and reliability of electronic systems, especially when precise and consistent conductance values are crucial for operation.

The intrinsic deviations in memristors can result in an imprecise deployment of a well-trained model on the corresponding synapse devices in one or more crossbar arrays. Correspondingly, the memristor conductances are no longer suitable to being treated as deterministic quantities. Therefore, we treat the weights as variational to make the weight distributions during the training process consistent with the programmed weights at memristor crossbars, thus compensating for the deviations through an adaptive training algorithm instead of post-calibration. This approach allows the model to capture and propagate the uncertainties in the weights and thus provides a more realistic representation of the model’s behavior in memristor crossbar arrays. More specifically, variation inference is employed to approximate the posterior weight distributions, providing computationally efficient techniques to incorporate memristor deviations into the training process.

To establish a unified Bayesian neural network framework for a fault-tolerant neuromorphic computing system, as discussed in Section 3, there are two crucial components that need to be modified:Log-normal variational posterior: We introduce a log-normal variational posterior to quantify and characterize the inevitable deviations in synapse weights represented by memristor crossbars, attempting to accommodate process variations and noise through algorithmic training.Variational Bayesian inference: We employ variational inference to find the network parameters of the log-normal variational posterior that minimize the KL divergence between the variational posterior and the true Bayesian posterior of weights.

For the underlying system that deploys neural network models on imprecise and random conductance states of memristors, we need a unified framework to incorporate the impact of device variations into algorithmic training. By employing a variation-aware prior distribution over the weight space that follows a log-normal distribution, the BNN enforces the posterior to better fit the true distribution of conductance states through iterative training, capable of adaptively tolerating to weight drifting. This can be achieved by minimizing the KL divergence between the two, i.e., KL[q(w|θ)||P(w)] in Equation (Equation 4). As expected, the proposed methodology unifies data observation with the fault tolerance capability, and is capable of finding an optimal weight posterior approximation in one training session. Besides the introduction of the log-normal variational posterior, the underlying learnt uncertainty in the weights of BNN models can be also used to improve the fault tolerance to random noise.

### 4.2. Log-Normal Variational Posterior

Generally speaking, deviations with different severities due to process variations are stochastically distributed in all the memristors in a crossbar array. Obviously, it is not enough to compensate their effects through retraining and/or remapping, which come at the huge hardware resources costs. It is therefore essential to develop a fault-aware learning algorithm to improve the robustness of NN models deployed on the hardware.

As discussed in Section 3, the Gaussian variational posterior is a typical case in Bayesian neural networks [24]. However, it is known that memristor device variation follows a log-normal distribution [11,12]. Specifically, the conductance of an on-state memristor can be expressed as $g\to e^{\lambda }\cdot g_{0}$, where $\lambda ∼N(0,\sigma^{2})$. $g_{0}$ is equivalent to the ideal value obtained from a well-trained NN model without considering any faults. σ is the scaling factor, denoting the severity of process variations. Having liberated the algorithm from the confines of Gaussian priors and posteriors, we employ a simple log-normal prior combined with a variation-aware log-normal posterior.

Suppose that the variational posterior is a log-normal distribution, i.e., ∼$\mathrm{LN}(\mu ,\sigma^{2})$. Then, a sample of the weights w can be obtained by sampling a unit log-normal distribution, shifting it by a mean μ and scaling by a standard deviation σ. In the context of memristor-based neuromorphic computing, the weights represented by memristor conductances deviate from the expected/ideal values $w_{0}$: $w\to e^{N(0,\sigma^{2})}\cdot w_{0}$. Thus, the synapse weight can be further expressed as a product of $w_{0}$ and a sample of a log-normal distribution with the mean as zero and the standard deviations as σ.

Moreover, the pointwise standard deviation is parameterized as σ=log(1+exp(ρ)) and is always non-negative. In the following, we represent the expected weights $w_{0}$ as μ, which are not the mean of the above log-normal distribution. Thus, a log-normal sample of the weights can be transformed from a sample of random noise ϵ and the variational posterior parameters θ=(μ,ρ) to:(5)$$w=\mu \cdot e^{\log(1+\exp(\rho ))\cdot \epsilon }$$
in which ϵ is sampled from a standard Gaussian distribution.

In this paper, we adopt the 1T1M-based memristor crossbar structure [28], where the relationship between weights and memristor conductances is g = α·w + β, where w is the expected weight vector learned from Equation (Equation 5) and g is the vector for the memristor conductances of the crossbar corresponding to w. After the variational posterior approximation is accomplished, the expected weight vector w can be mapped and programmed on the memristor crossbar arrays.

### 4.3. Variational Bayesian Inference

The reality is that the programming of a memristor is an inherently random process, so the devices are not well suited to being treated as deterministic quantities. In Bayesian neural networks, model parameters are not single high-precision values but probability distributions. Then, we can deploy variational posterior distributions that are trained with both variation-aware prior knowledge and data observations of the corresponding devices in crossbar arrays. In this way, a weight represented by the programmed conductance state falls within the variational posterior of tolerated error. When performing inference, the weights can be considered as being sampled from the posterior distributions, enabling a natural pairing of the algorithm and device variations.

Similar to Section 3, we here use P(w) as the priori to the BNN and q(w|θ) as the estimated weight posterior. Reviewing the available literature [23], only expressions for the KL divergence between two distributions from the same distribution family can be found. For the objective function in Equation (Equation 4), the complexity cost, i.e., the KL divergence term, needs to be derived with the prior and the variational posterior that are from the log-normal distribution family.

Without loss of generality, let the prior distribution $P∼\mathrm{LN}(\mu_{p},{\sigma_{p}}^{2})$ and the posterior $Q∼\mathrm{LN}(\mu_{q},{\sigma_{q}}^{2})$ with probability density functions *p* and *q*. Then, the KL divergence between two log-normal distributions is:(6)$$\begin{array}{cc} \mathrm{KL}[q(w|\theta )||P(w)] & =\int_{0}^{\infty }p\cdot \ln\left(\frac{p}{q}\right)=E\left[\ln\left(\frac{p(P)}{q(P)}\right)\right]\\ & =E\left[\ln\left(\frac{\sigma_{q}}{\sigma_{p}}\exp\left(\frac{{\left(\ln\left(P\right)-\mu_{q}\right)}^{2}}{2{\sigma_{q}}^{2}}-\frac{{\left(\ln\left(P\right)-\mu_{p}\right)}^{2}}{2{\sigma_{p}}^{2}}\right)\right)\right]\\ & =\ln\left(\frac{\sigma_{q}}{\sigma_{p}}\right)+\frac{1}{2{\sigma_{1}}^{2}}\left[{\left(\mu_{p}-\mu_{q}\right)}^{2}+{\sigma_{p}}^{2}-{\sigma_{q}}^{2}\right] \end{array}$$

For the posterior distribution q(w|θ) and the priori P(w) used in the proposed methodology, the mean values $\mu_{q}$ and $\mu_{q}$ of the above two log-normal distributions are zero; thus, the above formula can be simplified further:(7)$$\begin{array}{cc} \mathrm{KL}[q(w|\theta )||P(w)] & =\ln\left(\frac{\sigma_{q}}{\sigma_{p}}\right)+\frac{1}{2{\sigma_{1}}^{2}}\left({\sigma_{p}}^{2}-{\sigma_{q}}^{2}\right) \end{array}$$

Combined with the observed likelihood cost data, we can re-formulate the objective function of variational Bayesian inference. Each step of the optimization of the Bayesian neural network is as follows:
1.Sample ϵ∼N(0,1).2.Let the network parameters be $w=\mu \cdot e^{\log(1+\exp(\rho ))\cdot \epsilon }$, where μ denotes the expected weight values that are error-free.3.Let the variational parameters be θ=(μ,ρ).4.Let the objective function be $f\left(w,\theta \right)=\mathrm{KL}\left[q\left(w|\theta \right)||P\left(w\right)\right]-E_{q(w|\theta )}\left[\log P\left(D|w\right)\right]$.5.Calculate the gradient with respect to the parameter μ:
(8)$$\Delta_{\mu }=\frac{\partial f(w,\theta )}{\partial w}e^{\log(1+\exp(\rho ))\cdot \epsilon }+\frac{\partial f(w,\theta )}{\partial \mu }.$$6.Calculate the gradient with the standard deviation parameter ρ:
(9)$$\Delta_{\rho }=\frac{\partial f(w,\theta )}{\partial w}\frac{\mu \cdot \epsilon \cdot e^{\log(1+\exp(\rho ))\cdot \epsilon }}{1+\exp(-\rho )}+\frac{\partial f(w,\theta )}{\partial \rho }.$$7.Update the variational parameters μ and σ.

It is noted that the ∂f(w,θ)/∂w of the gradients $\Delta_{\mu }$ and $\Delta_{\rho }$ is shared and is exactly the gradients found by the usual backward propagation algorithm in a neural network. Then, the variational parameters can be learned from scaling and shifting the usual gradients as above. During forward/backward propagation, we adopt the minibatch technique for stochastic variational inference to average and propagate the gradients [26,29]. A typical local re-parameterization trick is also employed to save the computational cost in the variational Bayesian inference [26].

## 5. Evaluation

### 5.1. Experimental Setup

In our experiments, we utilize three commonly used neural network models, Multi-Layer Perceptron (MLP), LeNet, and AlexNet, and two widely used datasets: MNIST and CIFAR-10. These models represent different levels of complexity and have been widely employed in various applications. The classification accuracy is the measurement metric for our proposed methodology. Furthermore, the robustness with respect to the severity of variations can be also measured by the relative inference accuracy degradation between the cases with and without variations. The lower the degradation in the inference accuracy, the better the fault tolerance capability our methodology offers. The proposed Bayesian neural network framework with a log-normal prior and posterior distributions and its training setup were modified on the basis of the github repository (https://github.com/kumar-shridhar/PyTorch-BayesianCNN, accessed on 10 January 2021). The key innovations of this paper involve the modeling of the log-normal variational posterior and the derivation of the KL divergence between the variational posterior and the true posterior.

In the mentioned learning task, the training of BNN models is performed using the Adam [30] optimizer with specific hyperparameters (learning rate: 0.001, batch size: 64, epochs: 100). For the neuromorphic computing hardware, we adopted a similar setup to that in the literature [11], where each layer of the NN models can be mapped to one or more memristor crossbar arrays of size 64 × 64. By considering the hardware implementation using memristor crossbar arrays and accounting for the effects of process variation, the experiments aim to demonstrate the effectiveness of the proposed design methodology in achieving an efficient and robust classification performance.

### 5.2. Experimental Results

We first explore how the proposed methodology can compensate for deviations induced by process variations. Figure 3a,b presents a comparison of the inference performance of two neural network models, LeNet on MNIST and AlexNet on CIFAR-10, under different process variations. According to the relationship between the ideal weight and the programmed weight, i.e., $w\leftarrow w_{0}\cdot e^{N(0,\sigma^{2})}$, we use the standard deviation σ (e.g., the X-axes in the figures) to indicate the severity of deviations on the memristor conductances in the experiments.

Compared to the baselines (green lines in Figure 3), which suffer from a great loss in accuracy as σ increases, the proposed methodology can reduce the inference degradation for MNIST and CIFAR-10 classification tasks up to 47% and 42%, respectively. Thus, the execution models of memristor crossbar arrays trained using the BNN method present overwhelming advantages in terms of the inference performance, enabling more stable computational effectiveness against the uncertainties of process variations. In Figure 3a, under different levels of deviations (e.g., σ is increased from 0.1 to 0.4), the accuracy of MNIST remains stable at almost 0 inference degradation. This phenomenon is also present in the inference performance on CIFAR-10 in Figure 3b. By comparing Figure 3a with Figure 3b, AlexNet presents better fault tolerance than LeNet, in which the accuracy on CIFAR-10 can reach over 75% within σ=0.6. This indicates that the deeper network has a better healing capability and thus is more robust against uncertainties in the weights.

We further explore the tolerance of the proposed methodology to random noise. Even if the weights are well trained and process variations are accounted for, the procedures of mapping the intended conductances to devices and executing the inference may still suffer from a certain level of perturbation. Thus, both process variations and noise can contribute to the conductance deviations from the intended values, which eventually results in an inference accuracy degradation from the algorithmic level evaluation. Considering that both the read noise and write noise follow a Gaussian distribution [31], the actual weight is indeed a sum of the log-normal and the Gaussian noise, where the standard deviation of Gaussian noise was set to 5% of the nominal values based upon past experience [13].

We then comprehensively evaluated the overall robustness when suffering from the combined impacts of process variations and noise. Figure 4a,b compares the inference accuracy under different amounts of process variations as well as 5% Gaussian noise for the aforementioned two NN models. By employing the proposed methodology, both models can reduce the accuracy loss by up to ∼30% across the whole range of σ. However, for the baseline plotted in Figure 4a, when the variation effects are slightly larger, i.e., σ≥0.5, the accuracy decreases by 40%. Obviously, the effects of noise further degrade the accuracy by comparing Figure 4 with Figure 3. Impressively, a smaller accuracy gap, i.e., ∼5% for LeNet-MNIST, can be maintained when σ≤0.4 and the noise is within 5%, which is considered as a reliable fault tolerance range.

Finally, we compared the performance of the proposed methodology with other state-of-the-art methods [11]. Ref. [11] implemented a fault-tolerant framework, FTNNA, by replacing the original classifiers in NN models with a novel collaborative logistic classifier to improve its error-correction capability. This comparison allows for an assessment of the effectiveness and competitiveness of the proposed framework in relation to existing approaches. For a fair comparison, the setup of the proposed framework was consistent with [11], while the reported inference results of FTNNA were taken directly from the original paper. As shown in Table 1, it is clear that the proposed framework outperforms FTNNA by 4∼7% in terms of accuracy across the whole range of process variation σ, even without involving retraining/remapping.

## 6. Conclusions

This paper proposes a novel design methodology based on a Bayesian neural network to improve the fault tolerance of neuromorphic computing systems. The proposed methodology creatively combines the variational Bayesian inference technique with a fault-aware variational posterior distribution to accommodate the underlying deviations in memristors through one BNN training process. The experimental results show that the proposed framework can effectively rectify accuracy losses of up to ∼40% due to process variations and stochastic noise without involving expensive retraining and remapping.

## Figures and Tables

**Figure 1 micromachines-14-01840-f001:**
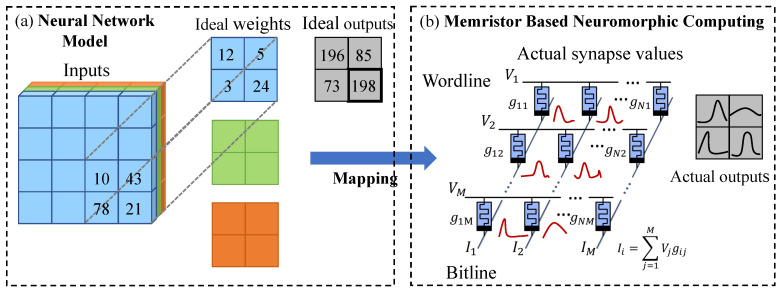
Performance deviation between (**a**) a well-trained neural network model and (**b**) its deployment on a memristor-based neuromorphic computing system.

**Figure 2 micromachines-14-01840-f002:**
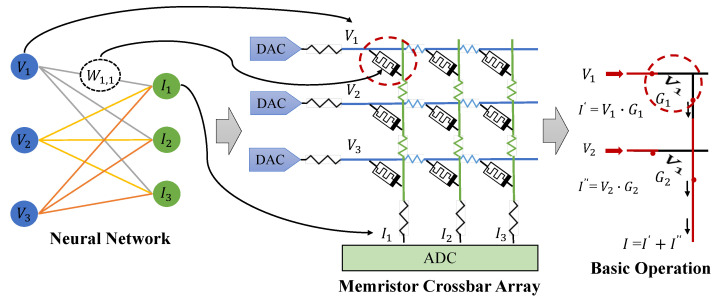
The matrix–vector multiplication (MVM) operation conducted on memristor crossbar arrays.

**Figure 3 micromachines-14-01840-f003:**
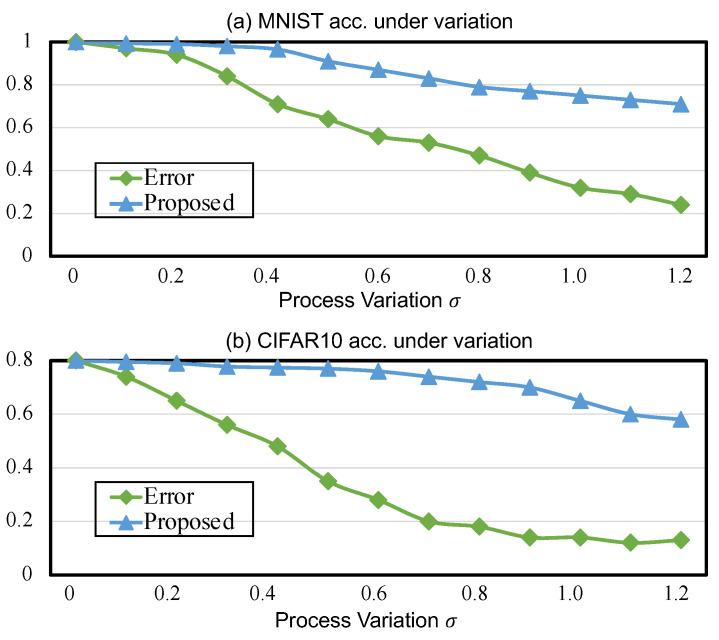
Comparison of inference accuracy under different process variations (σ) without/with the proposed methodology: (**a**) LeNet on MNIST and (**b**) AlexNet on CIFAR-10.

**Figure 4 micromachines-14-01840-f004:**
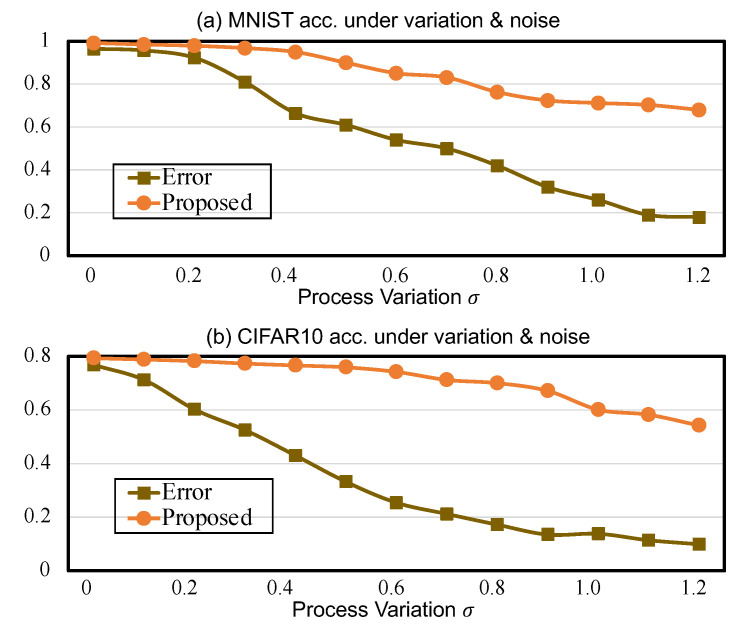
Comparison of inference accuracy under the combined effects of process variations (σ) and noise without/with the proposed methodology: (**a**) LeNet on MNIST and (**b**) AlexNet on CIFAR-10.

**Table 1 micromachines-14-01840-t001:** Comparison of inference accuracy for MLP-MNIST under different variations between the proposed methodology and a state-of-the-art method [11].

Method	Process Variation σ				
	0.2	0.4	0.6	0.8	1.0
FTNNA [11]	94%	90%	82%	77%	73%
Proposed	99%	97%	89%	81%	77%

## Data Availability

Not applicable.

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
