# Peer review of "A Design Methodology for Fault-Tolerant Neuromorphic Computing Using Bayesian Neural Network"

_micromachines, 2023, doi:10.3390/mi14101840_

Round 1

Reviewer 1 Report

Comments:

1.BNN is generally considered to alleviate overfitting problems and have some resistance to noise. Is there a reasonable explanation and proof for the resistance advantage of BNN to memristor variation and noise?

2.Is there a clear theoretical basis for assuming in Section 4.2 that the standard deviation sigma is non-negative, which means that memristor variation will only increase the weight rather than decrease it?

3.What are the specific advantages of using the method proposed in this article to modify a modified BNN compared to a regular BNN? It can be further explained with abaltion studies experiments.

4.This article compares the proposed method with FTNNA and Error, and display it in Figure 3, Figure 4, and Table 1. For better display, all methods can be drawn in the same figure. Additionally, it is strongly recommended to discuss related method. Regarding various other different methods, the following articles can be useful: 1)BayesFT: Bayesian Optimization for Fault Tolerance Neural Network Architecture: https://ieeexplore.ieee.org/document/9586115

2)Improving the robustness of analog deep neural networks through a Bayes-optimized noise injection approach https://www.nature.com/articles/s44172-023-00074-3

5.To enhance the persuasiveness of the method's universality, can it be applied not only to simple neural network architectures, but also to more complex ones (such as ResNet) ?

The quality of English is moderately good.

Author Response

Please see the attachment for easier reading.

1. The robustness study based on Bayesian neural network is a promising direction in the development of memristor-based neuromorphic computing. As the programming of resistive memory is an inherently random process, the devices are not well suited to being treated as deterministic quantities. Fortunately, such randomness follows stereotyped probability distributions, e.g., Gaussian or Lognormal distribution, allowing memristor to be instead yielded as physical random variables. In Bayesian neural networks, model parameters are not single high-precision values but probability distributions, suggesting a more natural pairing of algorithm and technology. In this setting, the objective is no longer to precisely transfer a single parameter from a software model to the corresponding device but to transfer a probability distribution from the software model into a distribution of device conductance states. 

Previous work has confirmed the robustness advantage of Bayesian inference for memristor-based inference hardware [1, 2]. By using the probability distributions that emerge from the volatile random switching properties memristor and stochastic electronic circuits as the prior knowledge of BNN training, the posterior distribution of each parameter encapsulates the uncertainty in its estimation. It means that the device conductance states are indeed sampled from the posterior of BNN. In the updated manuscript, we have modified Section 4.1 with more detailed explanation, highlighted in blue.

[1] Gao, D.; Huang, Q.; Zhang, G.L.; Yin, X.; Li, B.; Schlichtmann, U.; Zhuo, C. Bayesian inference based robust computing on memristor crossbar. In Proceedings of the 2021 58th ACM/IEEE Design Automation Conference (DAC). IEEE, 2021, pp. 1–6. 420.

[2] Dalgaty, T.; Esmanhotto, E.; Castellani, N.; Querlioz, D.; Vianello, E. Ex Situ Transfer of Bayesian Neural Networks to Resistive Memory-Based Inference Hardware. Advanced Intelligent Systems 2021, 3, 2000103. 

2. We apologize for the confusion. In Gaussian distribution and its variants, the expectation mu decides its position and the variance sigma^2 decides its amplitude. Whether the standard deviation sigma is positive or negative, the memristor conductance state is randomly sampled from the distribution governed by (mu, sigma^2), which may be less than or greater than the expectation value.

3. The main difference of the proposed BNN with a regular BNN is the selection of prior distribution. In a regular BNN, the prior distribution over the weights is often defined as Gaussian with the mean being zero and the variance being less than 1, that is mainly based on empirical results instead of early experiments. In this situation, the training sample size is much larger than the sample size of prior knowledge, so the prior knowledge can be ignored. 
For the proposed BNN, the posterior probability density is continuously updated with prior knowledge to analyze the probability distribution of parameters, as the prior probability follows the same distribution with the memristor conductance values.
Moreover, this paper mainly aims to confirm that the proposed BNN method can eliminate the accuracy degradation compared with the traditional frequentist inference approach. Therefore, a regular BNN model without any modification is not the reference baseline, so we do not display the corresponding ablation experiments in the manuscript.

4. Thanks for your advice. The reason we have not drawn the results of FTNNA in the figure is that we cannot reproduce this work and cannot acquire adequate and precise test data. To overcome it, we have no choice but to select several typical data points from the figure in the paper of FTNNA, which are used to roughly compare with our method. 

Prior work [3, 4] also employed Bayesian optimization method to improve the robustness of analog DNNs, presenting a thorough theoretical study about the interaction between fault tolerance and neural network architecture choice (i.e., dropout, normalization, etc.) as well as its influence on the robustness. Differently, Bayesian optimization in the above work [3, 4] is used to automatically search for the optimal neural network architecture robust to weight drifting, while the objective of our proposed BNN method is to learn the true posterior distribution presented by the memristor conductance states.
In the revised version, we have added the corresponding discussion in the last paragraph of Section 2, highlighted in blue. 

[3] Ye, N.; Mei, J.; Fang, Z.; Zhang, Y.; Zhang, Z.; Wu, H.; Liang, X. BayesFT: Bayesian Optimization for Fault Tolerant Neural Network Architecture. In Proceedings of the 2021 58th ACM/IEEE Design Automation Conference (DAC). IEEE, 2021, pp. 487–492. 
[4] Ye, N.; Cao, L.; Yang, L.; Zhang, Z.; Fang, Z.; Gu, Q.; Yang, G.Z. Improving the robustness of analog deep neural networks through a Bayes-optimized noise injection approach. Communications Engineering 2023, 2, 25.

5. Our methodology can be seamlessly extended to deeper models on more complex tasks (e.g., ResNet on CIFAR100). However, it takes a large amount of training resource to make the complex model reach convergence, so we cannot acquire adequate experimental data. Fortunately, the early experimental results indicate the comparison results between CNN and BNN in ResNet-CIFAR10 are similar to LeNet-MNIST and AlexNet-CIFAR10. It is worthy noting that this paper is mainly to prove that the proposed Bayesian framework on memristor crossbars can achieve performances superior to conventional training algorithm. In the future, we aim to further introduce more useful tricks to save the computation cost for higher optimization efficiency, promoting our methodology to real-word and large-scale scenarios. 

Reviewer 2 Report

Comments to Authors

The article entitled, “A Design Methodology for Fault-Tolerant Neuromorphic Computing using Bayesian Neural Network” discusses a present a novel design methodology for fault-tolerant neuromorphic computing utilizing Bayesian neural networks. This approach integrates the variational Bayesian inference technique with a fault-aware variational posterior distribution. The framework, grounded in Bayesian inference, effectively incorporates the influences of memristor deviations into algorithmic training. It optimizes weight distributions within neural networks to account for uncertainties and mitigate potential inference degradation. My decision is a moderate revision at this stage.

1.     Please elaborate on the specific types of process variations that memristor crossbar arrays are subject to and their potential impact on neuromorphic computing?

2.     What are the key challenges and limitations in achieving deterministic synapse weights in practical scenarios?

3.     Please provide more details on how the Bayesian neural network integrates variational Bayesian inference and a fault-aware variational posterior distribution to address the challenges posed by memristor deviations?

4.     In the context of algorithmic training, how does the proposed framework optimize weight distributions within neural networks to accommodate uncertainties? Please explain in more detail the mathematical or computational methods employed.

5.     What are the quantitative results or metrics used to assess the effectiveness of the fault-tolerant neuromorphic computing methodology, particularly in terms of its ability to handle process variation and noise?

6.     Are there any practical applications or real-world scenarios where this methodology has been tested and demonstrated to provide more robust computing on memristor crossbar arrays?

7.     How scalable is this methodology? Can it be applied to large-scale neuromorphic computing systems effectively?

8.     What are the potential implications or future directions for research and development in the field of neuromorphic computing based on the findings of this study?

9.     The present citations of this paper provide a very limited overview about memristors which is insufficient for the wide readership of journal. Authors are advised to add more recent literature related to memristors to give a wider overview about memristive devices by citing the following comprehensive review papers (doi of papers are given):

a.      https://doi.org/10.1080/14686996.2020.1730236

b.     https://doi.org/10.1021/acsaelm.1c00078

Author Response

Please see the attachment for easier reading.

  1. We appreciate the reviewer’s advice. Process variations are inherent from nano-technology manufacturing processes, and can be decomposed to global and local variations. Global components are shared by all the memristors on a die. Local variation is related to the proximity effect and results in variation correlation shared by the devices across the array. The neural network parameters with the combined impact of global and local variation deviate from the expected value, i.e., following a Lognormal distribution as w'=w e^{N(0,sigma^2)}. This inevitably hurts hardware inference performance, significantly deviating from the software validation accuracy. In the revised version, we have rewritten the fourth paragraph of Section 1 in Page 2 with detailed description about process variations.
  2. The key limitation in achieving deterministic synapse weights is the intrinsic cycle-to-cycle and device-to-device conductance variability. Such variability is also known as soft faults [5, 6], it means that the actual resistance of the memristor deviates from the targeted value, but the resistance can still be tuned. Soft faults are generally caused by variations associated with both fabrication techniques and write/read operations. To mitigate the impacts of the faults, the devices needs to being repeatedly programmed until their conductance falls within a disceretized window of tolerated error. Obviously, instead of correcting the synapse weights through such program-verify scheme, it is key to adaptively tolerate to stochastic behaviours on memristors by a more natural pairing of algorithm and technology. Inspired by this point, this paper proposes to leverage Bayesian neural network to incorporate the impact of memristor variations into algorithmic training.

    [5] Liu, M.; Xia, L.; Wang, Y.; Chakrabarty, K. Fault tolerance in neuromorphic computing systems. In Proceedings of the Proceedings of the 24th Asia and South Pacific Design Automation Conference, 2019, pp. 216–223.

    [6] Li, B.; Yan, B.; Liu, C.; Li, H. Build reliable and efficient neuromorphic design with memristor technology. In Proceedings of the Proceedings of the 24th Asia and South Pacific Design Automation Conference, 2019, pp. 224.

  3. Thanks for your advice. We have modified Section 4.1 on Page 5 with detailed information about how the proposed methodology integrates Bayesian inference and a fault-aware variational posterior.
  4. We appreciate the reviewer’s advice. In the updated manuscript, we have modified Section 4.3 on Page 6 with more explanation about the algorithmic training, highlighted in blue.
  5. We apologize for the confusion. We use the classification accuracy as the measurement metric for our proposed methodology. Furthermore, the robustness w.r.t the severity of variations can be also measured by the relative inference accuracy degradation between the cases with and without subject to variations. The original accuracies without considering device variations implemented with the selected neural network models under corresponding datasets serve as the upper bound of our fault-tolerant design. The less degradation of inference accuracy, the better fault tolerance capability our methodology offers. In the revised version, we have also added the following descriptions in the first paragraph of Section 5.1.
  6. We thank the reviewer for raising this important point. We have not make more ablation studies on practical applications or real-world scenarios due to the limitation of time. It is noted that this paper is mainly to prove that the proposed Bayesian framework on memristor crossbars can achieve performances superior to conventional training algorithm and state-of-the-art works. Therefore, the experiments are set up with identical NN models and tasks as prior works. 
  7. Our methodology can be seamlessly extended to deeper models on complex tasks (e.g. ResNet on CIFAR100). It takes a large amount of training resource to make large-scale model reach convergence, so we cannot acquire adequate experimental data to display in the manuscript. Fortunately, the early experimental results indicate the comparison results between CNN and BNN in ResNet-CIFAR10 are similar to LeNet-MNIST and AlexNet-CIFAR10. In the future, we aim to introduce more useful tricks to save the computation cost for higher optimization efficiency, further applying our methodology to large-scale or real-world neuromorphic computing system to verify its effectiveness.
